# Functional Relevance of the Long Intergenic Non-Coding RNA Regulator of Reprogramming (Linc-ROR) in Cancer Proliferation, Metastasis, and Drug Resistance

**DOI:** 10.3390/ncrna9010012

**Published:** 2023-01-31

**Authors:** José A. Peña-Flores, Diego Enríquez-Espinoza, Daniela Muela-Campos, Alexis Álvarez-Ramírez, Angel Sáenz, Andrés A. Barraza-Gómez, Kenia Bravo, Marvin E. Estrada-Macías, Karla González-Alvarado

**Affiliations:** Faculty of Dentistry, Universidad Autónoma de Chihuahua, Chihuahua 31000, Mexico

**Keywords:** linc-ROR, lincRNA-ROR, lncRNA, cancer proliferation, cancer progression, cancer metastasis, cancer invasion, drug resistance

## Abstract

Cancer is responsible for more than 10 million deaths every year. Metastasis and drug resistance lead to a poor survival rate and are a major therapeutic challenge. Substantial evidence demonstrates that an increasing number of long non-coding RNAs are dysregulated in cancer, including the long intergenic non-coding RNA, regulator of reprogramming (linc-ROR), which mostly exerts its role as an onco-lncRNA acting as a competing endogenous RNA that sequesters micro RNAs. Although the properties of linc-ROR in relation to some cancers have been reviewed in the past, active research appends evidence constantly to a better comprehension of the role of linc-ROR in different stages of cancer. Moreover, the molecular details and some recent papers have been omitted or partially reported, thus the importance of this review aimed to contribute to the up-to-date understanding of linc-ROR and its implication in cancer tumorigenesis, progression, metastasis, and chemoresistance. As the involvement of linc-ROR in cancer is elucidated, an improvement in diagnostic and prognostic tools could promote and advance in targeted and specific therapies in precision oncology.

## 1. Introduction

Cancer is a group of multifactorial diseases responsible for at least 10 million deaths around the world in 2020 alone [1]. Human cancer diversity exceeds the 200 types, observing clear differences among the origin of cells, acquisition of somatic mutations, variability in altered transcription pathways, and influences in the microenvironment of local tissues [2]. As cancer advances, new mutations produce a greater genetic heterogeneity to form the primary tumor, eventually eroding the basal membrane and spreading to other regions via the circulatory system [3,4]. This event is called metastasis and proposes a challenge to scientists and clinicians since its occurrence leads to a high recurrence and poor survival rate. Chemotherapy is a very common treatment for cancer patients, but unfortunately, most tumors exert drug resistance, resulting in around 90% of deaths in cancer patients [5]. Epigenetic processes such as DNA methylation, histone acetylation, and lncRNA interaction regulate drug transporters and metabolic enzymes, thus promoting cancer chemoresistance [6].

A vast collection of evidence shows that only 2% of the transcribed human genome codes for proteins, whereas the remaining 98% of RNAs are non-coding [7,8]. Recent advances in sequencing technology have shifted the prior assumption that non-coding RNA (ncRNA) was a “junk” transcriptional product to transcripts comprising signals that control gene expression and are essential both in normal physiological function and in disease [8,9].

Since the discovery of small regulatory ncRNAs in the 1990s, substantial progress has been made to catalog a rapidly increasing number of both short and long ncRNAs [10]. MicroRNA (miRNA), long non-coding RNA (lncRNA), circular RNA (circRNA), and PIWI interacting RNA (piRNA) are the four major ncRNA types with distinct functions in cancer [11]. While miRNAs usually bind to targeted mRNA to degrade it [12,13], lncRNAs regulate gene expression by exerting multiple mechanisms including the recruitment of polymerase II and diverse transcription factors [14,15], regulating alternative splicing of pre-mRNAs [16,17], sequestering miRNAs to prevent them from performing their function [18,19,20] or acting as a scaffold on protein–protein interactions [21,22,23].

Increasing documentation suggests that linc-ROR, a long intergenic non-coding RNA, is implicated in cancer proliferation, metastasis, and drug resistance [24,25,26]. On this matter, this review aims to contribute to the current comprehension of linc-ROR and its implication in cancer proliferation, metastasis, and drug resistance. The review provides up-to-date synthesized information on linc-ROR targets, their relationship with other components, and the effects reported by a plethora of researchers. A systematic search was performed on PubMed, Google Scholar, Cochrane Library, Web of Science, and EMBASE up to January 2023 for articles matching the following criteria: (long intergenic non-coding RNA regulator of reprogramming (linc-ROR), or lincRNA-ROR, or lncRNA-ROR and cancer proliferation, or cancer metastasis, or cancer invasion, or cancer migration, or cancer progression, or cancer drug resistance). The titles and abstracts were screened and the relevant full-text manuscripts were acquired for further analysis.

## 2. Overview of Long Non-Coding RNAs

Long non-coding RNAs (lncRNAs), also known as competing endogenous RNAs (ceRNAs) [27], are a heterogeneous RNA family that comprise transcripts of 200 nucleotides or longer and are coded in the genome but not translated into proteins [28,29]. According to GENCODE, more than 56,000 lncRNA transcripts in almost 19,000 genes have been identified in humans [30]. Recent evidence suggests that lncRNAs are implicated in several cancer progression mechanisms including proliferation [31,32], differentiation [33,34], autophagy [35,36], epithelial–mesenchymal transition (EMT) [37], invasion [38,39,40], and metastasis [41,42]. LncRNAs are often found as modulators of signaling cascades at the epigenetic, transcriptional, posttranscriptional, translational, or posttranslational levels [43]. Cancer-controlling lncRNAs are classified as proto-oncogenic or tumor suppressors based on their function, being the tumorigenic lncRNAs expressed as cancer drivers that activate the cell cycle and exert anti-apoptosis effects [44]. On the other hand, tumor suppressors are generally downregulated in tumor biopsies, and evidence suggests that overexpression of these lncRNAs halts some of the cancer mechanisms [43].

Based on their structural origin and relative position to protein-coding genes, lncRNAs can be classified as (a) divergent (pancRNA) when they originate from the same promoter region as the protein-coding gene, but from the opposite strand; (b) convergent when genes are encoded on opposite strands, facing each other and convergently transcribed; (c) overlapping when genes extend along the same or opposite strand; (d) enhancer RNAs expressed as uni- or bidirectional transcripts; (e) intronic, when transcribed from an intron of another gene; (f) host lncRNA for miRNA; and (g) intergenic lncRNA (lincRNA) when the transcript is located distant from other genes [45,46]. Additionally, covalently closed circular RNAs (circRNAs) are produced by the back splicing of exons, requiring spliceosomal machinery for their biogenesis [47].

Over the last few years, substantial advances in RNA sequencing have allowed the identification of both physiological and pathological involvement of lncRNAs through four basic mechanisms: signal, decoy, guide, and scaffold [48,49,50]. Some lncRNAs function as signals to regulate the initiation, elongation, or termination of actions by transcription factors [51]. Other lncRNAs function as decoys by binding to transcription protein complexes to deviate from their target DNA [50]. Most lncRNAs have been related to act as molecular sinks for miRNAs, mediating gene expression by acting on splicing regulators and other genetic and epigenetic components [52].

In diabetes, the overexpression of lncRNA maternally expressed gene 3 (MEG3) was shown to suppress endothelial–mesenchymal transition (endMT) in diabetes retinopathy through inhibition of the PI3K/Akt/mTOR signaling pathway [53]; similarly, lncRNA H19 overexpression prevented glucose-induced endMT in human retinal endothelial cells [54]. The knockdown of lncRNA myocardial infarction-associated transcript (MIAT) decreased the proliferation and migration of cultured human carotid artery smooth muscle cells (SMCs) through the regulation of the EGR1-ELK1-ERK pathway in atherosclerosis and carotid artery disease [55]; Ye et al. [56] reported MIAT as a miR-149-5p sponge to positively modulate the expression of anti-phagocytic molecule CD47, inhibiting efferocytosis in advanced atherosclerosis. Regarding neurodegenerative diseases, significant advances have been made in the identification of novel lncRNAs and their involvement in disease etiology and progression. In Parkinson’s disease (PD) mice, the lncRNA metastasis-associated lung adenocarcinoma transcript 1 (MALAT1) was highly expressed and promoted neuroinflammation through inducing inflammasome activation and reactive oxygen species (ROS) production [57]. In another study, animal experiments suggested that lncRNA taurine up-regulated 1 (TUG1) downregulation significantly improved the motor coordination ability of PD mice and inhibited the expression of inflammatory factors [58]; correspondingly, TUG1 expression was significantly upregulated in synovial fibroblast-like synoviocytes, activating invasion, migration, glucose metabolism, and inhibited apoptosis via miR-34a-5p interaction in rheumatoid arthritis [59], reinforcing the results of previous studies where its expression induces the production of inflammatory factors.

A plethora of studies have shown the dysregulation of many lncRNAs in cancer, often found as regulators in tumorigenesis, progression, metastasis, and drug resistance by modulating signaling cascades in many transcriptional and translational levels [60]. For instance, lncRNA HOXC-AS3 was found to mediate the oncogenesis of gastric cancer by the activation of abnormal histone modification [61]; in a similar fashion, Su et al. [62] found the same transcript overexpressed in human non-small-cell lung cancer specimens and cells, promoting growth and metastasis. The novel lncRNA UPLA1 (upregulation promoting LUAD-associated transcript-1) was highly expressed in the nucleus of lung adenocarcinoma cells, significantly improving the growth of tumors by promoting the Wnt/β-catenin signaling pathway [63]. Another novel lncRNA, uc.134, was found to be downregulated in hepatocellular carcinoma tissue samples, repressing cancer progression by inhibiting the CUL4A-mediated ubiquitination of LATS1 and increasing YAP^S127^ phosphorylation [64].

## 3. Linc-ROR in Disease

Long intergenic non-coding RNAs (lincRNAs) are RNAs autonomously transcribed that do not overlap annotated coding genes [65]. The long intergenic non-coding RNA, regulator of reprogramming (linc-ROR) consists of a four exon-long transcript with a length of 2603 nucleotides, localized in chromosome 18q21.31 and identified by chromatin lysine 4 and lysine 36 marks [66]. Most of the sequence is composed of long and short interspersed retro transposable elements (LINEs and SINEs along with long terminal repeats (LTRs)) called retrotransposons [67]. There is evidence that linc-ROR acts as a molecular sink for many miRNAs with many potential binding sites demonstrated by bioinformatic tool analysis. For instance, miR-145 complementarily binds to the ROR sequence between 2055 bp and 2059 bp [68]; for miR-205-5p, a binding site at ROR 791–810 bp sequence and 317–345 bp sequence for miR-34a-5p has been reported [69]. The binding between linc-ROR and miR-194-3p was determined via a luciferase reporter gene assay with two binding sites predicted by DIANA-LncBase, the first at 1906–1923 bp, and the second at 2378–2389 bp [70]. Many other reports have predicted different binding sites for miRNAs, positioning linc-ROR as a competing endogenous RNA.

Linc-ROR was first identified in pluripotent stem cells as a ceRNA of miRNAs involved in core transcription factors regulatory circuitry [71]. In addition, Zou et al. [72] demonstrated that linc-ROR maintained SOX2 gene expression through competitive binding to miR-145, achieving pluripotency maintenance in human amniotic epithelial stem cells. Another study showed that linc-ROR is downregulated in osteoporosis by inhibiting osteoblast proliferation via targeting miR-145-5p, highlighting its positive correlation in cell proliferation and stemming [73]. Similarly, Feng et al. [74] studied the role of linc-ROR in bone marrow mesenchymal stem cell chondrogenesis and cartilage formation; the results revealed that linc-ROR functioned as a miRNA sponge for miR-138 and miR-145, activating SOX9 expression and chondrogenesis activity. Linc-ROR has also been documented as an angiogenesis promoter through the downregulation of miR-26 and activation of NF-kappa B and JAK1/STAT2 signaling pathways [75].

Little research has been made concerning the role of linc-ROR in cardiovascular diseases. In a study on the crosstalk between cardiac microvascular endothelial cells (CMECs) and cardiomyocytes (CMs), it was found that linc-ROR downregulated its target miR-145-5p leading to activation of the endothelial nitric oxide synthase (eNOS) pathway, therefore increasing the survival rate of both CMECs and CMs [76]. Another research found a significant upregulation of linc-ROR in a hypoxia/reoxygenation (H/R) injury model, acting as a sponge for miR-138, aggravating H/R-induced myocardial cell injury [77]. In a viral myocarditis cell model, linc-ROR destroyed the mRNA stability of Forkhead Box P Factor 1 (FOXP1) by binding polypyrimidine tract binding protein 1 (PTBP1), promoting coxsackievirus B3-induced cardiomyocyte inflammation [78].

Regarding the relationship of linc-ROR with mental and neurodegenerative diseases, little evidence has also been reported. Tamiskar et al. [79] measured expression levels of several lncRNAs in the circulation of Parkinson’s disease patients to try to establish a possible correlation; linc-ROR was higher in PD patients compared with controls, revealing linc-ROR dysregulation. A similar study in schizophrenia patients revealed the presence of a sex-based dysregulation of lncRNAs when compared with healthy subjects, with linc-ROR upregulated and correlated with age [80]. Moreover, linc-ROR and other seven lncRNAs were found upregulated in the circulation or post-mortem brain tissues of schizophrenia patients [81]. Conversely, Hasemian et al. [82] also quantified expression levels of lncRNAs in the peripheral blood of epileptic patients and found no significant difference in the expression of linc-ROR between patients and controls. In a different study, the p53 regulatory pathway was correlated with linc-ROR upregulation in ischemia-induced apoptosis, exerting a combined effect on ischemic stroke recurrence [83]. Similarly, expression of linc-ROR increased significantly in middle cerebral artery occlusion in mice and it also promoted ASK-1/STRAP/14-3-3 complex formation to inhibit the activation of TNF-α/ASK-1-mediated apoptosis of human brain microvascular endothelial cells, indicating a potential role in cerebral hypoxia-induced injury [84].

The involvement of linc-ROR in cancer proliferation and metastasis has been documented, suggesting an important role in the clinicopathological characteristics of tumors, therefore considered an oncogene that affects prognosis, survival rate, and higher recurrence rate [85]. Intriguingly, a few studies have positioned linc-ROR as a tumor-suppressor lncRNA, proposing it could be involved in several mechanistic pathways exerting multiple and even opposite functions [86,87,88]. The aim of this review is to analyze and discuss the current state of linc-ROR research in the proliferation, metastasis, and drug resistance of human cancer. Functional and regulatory mechanisms of linc-ROR in distinct types of cancer are summarized in Table 1.

## 4. Linc-ROR in Cancer Proliferation and Progression

Cancer is considered a heterogeneous group of diseases characterized by the uncontrolled growth of cells and invasion of other tissues [175,176]. It has been suggested that cancer initiation is marked by the overexpression of proto-oncogenes, downregulation of tumor-suppressing genes, and halting of the DNA repairing machinery in a single cell [177,178,179]. Recently, more than 100 lncRNAs have been found to be regulators of tumorigenesis and progression, highly expressed as cell proliferation promotors and exerting anti-apoptosis effects [43,44]. In contrast, other lncRNAs such as the growth arrest-specific transcript 5 (GAS5), the maternally expressed gene 3 (MEG3), and the NF-kB interacting lncRNA (NKILA) function as tumor suppressors, inhibiting proliferation and tumor growth when up-regulated [43,180,181,182].

In this regard, many efforts have been made to establish the role of linc-ROR in cancer proliferation and progression. One of the most studied mechanisms by which linc-ROR promotes cell proliferation and cancer progression is by acting as a sponge for miRNA-145, a micro RNA considered a tumor suppressor in diverse types of cancers [183]. For instance, linc-ROR and p53 were upregulated in non-small-cell lung cancer (NSCLC) samples, whereas miR-145 was downregulated promoting proliferation, migration, and invasion in an NSCLC cell line [121]. Another study with pancreatic cancer samples and cell lines demonstrated that linc-ROR activates the derepression of core transcription factor Nanog by acting as a ceRNA and becoming a sink for miR-145 [128]. Similarly, Wang et al. [142] analyzed esophageal squamous cell cancer (ESCC) samples and cell lines to reveal that linc-ROR modulates the derepression of SOX9 by directly sponging multiple miRNAs, including miR-145. Another experiment confirmed linc-ROR upregulation along with the lamin B2 (LMNB2) gene in ESCC, whereas miR-145 was capable of reversing the proliferation and migration of ESCC cells previously promoted by ROR [146]. In a study involving endometrial carcinoma samples, linc-ROR was associated with the pluripotent state of endometrial tumors, partially by acting as a miR-145 sponge to inhibit mediation of the differentiation of cells, therefore promoting progression, stemness, and aggressiveness [108]. In another experiment involving stem cells in colorectal cancer (CRC), it was found that linc-ROR functions as a key ceRNA to prevent core transcription factors from miR-145-mediated suppression, regulating cell proliferation [147]. Arunkumar et al. [128] identified linc-ROR, c-Myc, KLF4, OCT4, and SOX2 overexpression in 60 oral squamous cell carcinoma tissue samples along with miR-145-5p downregulation, suggesting the existence of a ceRNA network in certain undifferentiated oral tumors. In a different approach aiming to demonstrate the presence of linc-ROR in intercellular exosomes, Takahashi et al. [128] demonstrated that linc-ROR is a hypoxia-responsive lncRNA in hepatocellular cancer (HCC) by sponging miR-145, generating intercellular signaling to promote cell survival during hypoxic stress.

Breast and colorectal cancers are two of the most prevalent cancers worldwide [1], which has driven investigators to try to find the relationship between linc-ROR and the proliferation and progression of these cancers. Estrogen receptor positive (ER+) breast cancer cell lines were utilized to knock out linc-ROR expression through CRISPR/Cas9 technology, revealing that linc-ROR functions as an onco-lncRNA to promote growth through activation of the MAPK/ERK pathway by regulating ERK-specific phosphatase DUSP7 [89]. Another study silenced linc-ROR, which inhibited breast cancer progression via repression of transmethylase MILL1 and TIMP3 [97], while Wnt/β-catenin pathway was blocked through linc-ROR downregulation in MDA-MB-231 breast cancer cells, decreasing colony and sphere numbers along with viability, migration, and invasion inhibition of breast cancer stem cells [102]. Similarly, Hou et al. [101] denoted that overexpression of linc-ROR seems to be responsible for promoting the proliferation and invasion of cancer cells as well as tumor growth by upregulating critical favors in the TGF-β signaling pathway. Another research performed in both breast and colorectal cancer found that linc-ROR increases c-Myc whilst interacting with the heterogenous nuclear ribonucleoprotein 1 (hnRNP1) and the AU-rich element RNA-binding protein 1 (AUF1), promoting cell proliferation and tumorigenesis [96]. A positive correlation was found between linc-ROR and epidermal growth factor receptor (EGFR) signaling in colon cancer, in which linc-ROR served as a tumor-promoting factor via repressing the ubiquitination and degradation of EGFR signaling [151]. Comparably, Li et al. [152] and Li et al. [153] established a correlation between linc-ROR overexpression and increased cell proliferation and viability in CRC tumor tissues through different pathways, positioning linc-ROR as an onco-lncRNA.

Some of the molecular regulatory mechanisms of linc-ROR in the progression and proliferation of gastrointestinal tract cancers have also been documented. A study with gastric cancer samples and cell lines revealed that linc-ROR led to the upregulation of several key stemness transcriptional factors such as OCT4, SOX2, Nanog, and CD133, promoting proliferation and invasion of gastric cancer stem cells [114]. A probable function interaction between the spalt-like transcription factor 4 (SALL4) and linc-ROR was associated with tumor maintenance and aggressiveness in gastric cancer (GC) tissues with helicobacter pylori infection [110]. Mi et al. [112] worked on GC cell lines to relate the linc-ROR upregulation with the miR-212-3p/FGF7 axis, promoting proliferative, migratory, and invasive capabilities of GC cells. In a study with hepatocellular cancer (HCC) cell lines and xenografts, linc-ROR acted as a ceRNA for miR-130a-3p to stabilize DEP domain containing 1 (DEPDC1) mRNA, facilitating progression and angiogenesis [125]. Moreover, linc-ROR functioned as a sponge for miR-223-3p decreasing the expression of tumor suppressor gene NF2, promoting cell proliferation and invasion [130]. Furthermore, linc-ROR acted as an miRNA sink to several tumor suppressor miRNA members of the let-7 family in pancreatic cancer, promoting cell migration and invasion [134]. The Hippo/YAP pathway was activated by linc-ROR in pancreatic cancer cell lines with increased proliferation, migration, and invasion [24]. Interestingly, HepG2 HCC cells were able to transfer its linc-ROR to normal liver cells via exosomes, also promoting proliferation and suppressing apoptosis [131]. Gao et al. [144] studied ESCC tumor samples and found a positive correlation with the expression of MDM2 by acting as a molecular sponge of miR-204-5p, modulating cell apoptosis and regulating p53 ubiquitination. Contrary to most of the research, Bai et al. [117] found a decline in linc-ROR expression in gastric cancer tissues and cell lines, acting as a tumor suppressor to restrain progression via the miR-580-3p/ANXA10 pathway.

Several studies in kidney and bladder cancer agree in positioning linc-ROR as a tumor promoter through different pathways and molecular functions. According to Shi et al. [159], linc-ROR binds to miR-206 to induce cell proliferation by regulating VEGF in renal cancer cell lines Caki-1 and Caki-2. Comparably, linc-ROR served as a decoy oncoRNA that blocked binding surfaces, preventing the recruitment of histone-modifying enzymes, thereby specifying a new pattern of histone modifications that promote tumorigenesis in renal cancer 293T cells [160]. In bladder cancer tissue samples and cell lines, ZEB1 was a target gene of linc-ROR, promoting tumor cell proliferation and inhibition of cell apoptosis [157]. In osteosarcoma, linc-ROR was a growth and metastasis promoter via modulating YAP1, the target gene of miR-185-3p, demonstrating its ability to act as a molecular sink for microRNAs [171]. Moreover, miR-206 was verified to be a target of linc-ROR in osteosarcoma tissue samples, whereas elevated linc-ROR was closely correlated with advanced tumor–node–metastasis (TNM) stage, lymph node metastasis, and poor overall survival rate.

Several regulatory pathways involving linc-ROR have been identified in head and neck cancer. For instance, the p53 gene expression was negatively correlated to linc-ROR in two studies involving nasopharyngeal carcinoma (NPC); in both cases, linc-ROR acted as a tumor growth promotor and apoptosis suppressor by inhibiting the p53 pathway [139,140]. In tongue squamous cell carcinoma (TSCC), linc-ROR was highly expressed, upregulating the expression of the LIM domain only 4 (LMO4) gene to promote the activation of the AKT/PI2K signaling pathway, stimulating cell proliferation and invasion [137]. Zhang et al. [138] compared serum samples from NPC patients and healthy donors to discover that linc-ROR was substantially expressed in serum exosomes from the NPC patients, increasing proliferation, migration, and angiogenesis through the AKT/VEGFR2 pathway. Conversely, the linc-ROR expression in glioblastoma tumor cells was significantly lower than in normal glial cells, acting as a tumor-suppressor lncRNA by inhibiting the expression of a key component to mTORC2 named Rictor, which in turn suppressed the AKT pathway activity and impaired the expression of glycolytic effectors including Glut1, HK2, PKM2, and LDHA [168]. Moreover, the expression of linc-ROR inhibited the proliferation of cancer cells and self-renewal of glioblastoma, partly by inhibiting the KLF4 expression in glioma U87 cell line [87]. Similarly, linc-ROR was significantly downregulated during progression from normal, hyperplastic, and adenomatous parathyroid to parathyroid carcinomas, concluding that linc-ROR may function as a tumor suppressor during parathyroid tumor progression [164]. Some of the molecular pathways and mechanisms regarding linc-ROR involvement in cancer proliferation and progression are depicted in Figure 1.

## 5. Linc-ROR in EMT, Cancer Invasion and Metastasis

Epithelial-to-mesenchymal transition (EMT) is defined as a highly coordinated biological process where epithelial cells lose adhesion, gain enhanced migratory capacity, and invade the extracellular matrix to become mesenchymal cells [184,185]. Once expressing mesenchymal signaling, cells can conveniently travel to distant suitable niches to metastasize and re-express epithelial characteristics in a reversed process called mesenchymal-to-epithelial transition (MET) [186]. This system has been extensively documented as an important process in development and in some diseases, mainly cancer metastasis [187,188]. In breast cancer, linc-ROR expression both in plasma and in tumor tissues was positively correlated to lymph node metastasis and estrogen and progesterone receptors [93,94], while similar results were obtained in high-grade ovarian serous cancer [105]. In colorectal cancer samples, linc-ROR and the oncogenic lncRNA colon cancer-associated transcript 1 (CCAT1) were upregulated and significantly associated with synchronous metastases [155]. Similarly, linc-ROR high expression was associated with poor prognosis in terms of tumor undifferentiation, lymph node infiltration, and postoperative recurrence of renal cancer [156], whereas exosomes transferred linc-ROR to induce EMT and inculcate the local tumor microenvironment and the distant metastatic niche in thyroid papillary carcinoma cell line TPC-1 [163].

Many molecular pathways have been related to EMT, some of which reveal a relationship with linc-ROR expression. For instance, the E-cadherin transcriptional repressors zinc finger e-box binding homeobox 1 (ZEB1) and 2 (ZEB2) have been implicated in EMT and tumor metastasis [189]. In two breast cancer studies, the linc-ROR acted as a miR-205 sponge to overexpress ZEB1 and ZEB2 and decrease the expression of E-cadherin [69,92]. E-cadherin membrane localization was impacted by the upregulation of Mucin1 (MUC1) through linc-ROR function as a ceRNA for miR-145 in triple-negative breast cancer (TNBC), increasing invasion and metastasis in a breast cancer cell line [100]. Similarly, Chen et al. [157] demonstrated that linc-ROR overexpression facilitated cell metastasis and contributed to the formation of an EMT phenotype in bladder cancer. Another explored axis that promotes EMT in gastric cancer involved the sequestering of miR-212-3p by linc-ROR to express the fibroblast growth factor 7 (FGF7), contributing to promoting migratory and invasive capabilities of gastric cancer cell lines [112]. Moreover, E-cadherin was decreased during linc-ROR expression, also favoring EMT activation through increasing expression of vimentin [113]. In several hepatocellular lines, the linc-ROR knockdown notably suppressed EMT by downregulating twist family bHLH transcription factor 1 (TWIST1) expression, promoting vimentin, and blocking E-cadherin [124]. Sun et al. [133] examined pancreatic cancer serum samples from patients, demonstrating that linc-ROR is delivered via exosomes to dedifferentiate adipocytes in the tumor microenvironment, inducing EMT via the hypoxia-inducible factor 1α (HIF1α), thus overexpressing ZEB1.

According to many studies, the function of linc-ROR as a molecular sink for tumor-suppressor miR-145 promotes EMT and cancer invasiveness capacity. In a tissue microarray study of papillary thyroid cancer samples, the silencing of linc-ROR increased the expression of miR-145, supporting the role of ROR as an endogenous miR-145 sponge [162]. Linc-ROR was dramatically upregulated in TNBC and in metastatic disease, apparently through regulation of the miR-145/ARF6 (ADP ribosylation factor 6) axis [99]. The filamin B (FLNB) gene was proved to be downregulated by linc-ROR by acting as a sponge for miR-145 in ovarian cancer [107]. The same linc-ROR/miR-145 axis overexpressed the fascin-acting-bundling protein 1 (FSCN1) gene, known to increase cell motility and directly related to EMT in lung adenocarcinoma cell lines [118]. The same axis was explored by Shang et al. [145] in ESCC tissues and cell lines, reaching the same conclusions. A further gene proved to be activated by miR-145 downregulation is ZEB2, which induced EMT and metastasis in hepatocellular cancer [25].

The Wnt/β-catenin pathway has been subject to extensive research owing to its crucial roles in tumorigenesis as a cancer stem cell renewal and cell proliferation promoter [190]. In this regard, a breast cancer study demonstrated that linc-ROR overexpression increased stem property and EMT process by exerting a positive correlation in the Wnt/β-catenin pathway [102]. Similarly, linc-ROR induced EMT in ovarian cancer cells by activating the Wnt/β-catenin cascade [104].

Other molecular pathways and mechanisms have been explored on the influence that exerts linc-ROR in EMT and metastasis. Jin et al. [111] demonstrated that linc-ROR knockdown downregulated the high mobility group AT-hook 2 (HMGA2) gene and upregulated miR-519d-3p to restrain migration, invasion, and EMT in gastric cancer, establishing a correlation with the miR-519d-3p/HMGA2 network. Similarly, the EMT phenotype induced by TGF-β1 was reversed after linc-ROR was knocked out in gallbladder cancer cells, whilst linc-ROR expression level was significantly associated with tumor sizes and lymph node metastasis [191]. In pancreatic cancer, linc-ROR promoted migration and invasion by activating the Hippo/YAP pathway, suggesting that YAP might be an underlying target to mediate EMT [24]. The p53 signaling pathway was suppressed in NPC by linc-ROR overexpression, promoting metastasis, apoptosis, and chemoresistance [140]. Moreover, a renal cancer study indicated that linc-ROR promotes proliferation and metastasis by binding to miR-206 and promoting VEGF overexpression [159]. Current studied molecular processes involving linc-ROR with cancer invasion, EMT, and metastasis are illustrated in Figure 2.

## 6. Linc-ROR in Drug Resistance

Nowadays, chemotherapy resistance is one of the most studied fields in cancer research since some studies suggest that it is responsible for up to 90% of deaths in patients receiving some type of drug therapy, becoming a concern for clinicians and researchers [5]. Epigenetic processes such as DNA methylation, histone acetylation, and ncRNA interaction have been evidenced to influence the expression of genes known to be active during drug resistance mechanisms [192]. The participation of lncRNAs in chemoresistance has been extensively studied in recent years, demonstrating the existence of novel drug resistance promotors and suppressors through many different pathways [36,193,194]. For instance, the small nucleolar RNA host gene 15 (SNHG15) lncRNA was induced by Sox12 to promote chemoresistance in cervical cancer by sponging miR-4735-3p [195], whereas the colorectal neoplasia differentially expressed (CRNDE) lncRNA was found to participate in autophagy regulation, attenuating chemoresistance in gastric cancer via the SRSF6-regulated alternative splicing of the phosphatidylinositol binding clathrin assembly protein (PICALM) [196].

While further investigation is required to reinforce knowledge of the active role of linc-ROR in chemoresistance, some studies have made important contributions. As previously documented, miR-145 has been extensively related to tumor activity suppression by several mechanisms, including augmented drug sensitivity [197,198]. In addition, linc-ROR is reported as a ceRNA for miR-145, therefore exerting a drug-resistance role in tumor cells of various types of cancer. Pan et al. [118] demonstrated the acquisition of chemoresistance and EMT phenotypes of docetaxel-resistant lung adenocarcinoma cells when miR-145 was sequestered by linc-ROR, enabling FSCN1 expression. In esophageal cancer tissue samples and cell lines, linc-ROR modulated the derepression of SOX9 by directly sponging multiple miRNAs, including miR-145, contributing to cell motility, chemoresistance, and self-renewal capacity [142]. Comparably, two colorectal cancer studies demonstrated the same correlation with negative regulation of miR-145 and thus decreasing sensitivity to chemo- and radiotherapy [147,149]. Moreover, linc-ROR functioned as a molecular sink for miR-145 to regulate the lncRNA RAD18 E3 ubiquitin-protein ligase (RAD18) expression, promoting DNA repair and radioresistance in hepatocellular cancer cell lines [122].

Cisplatin resistance has been associated with linc-ROR upregulation in cancer cells. In gastric cancer cell lines MKN45 and HGC-27, linc-ROR acted in the miR-519-d-3p/HMGA2 network to promote cisplatin resistance [111], whilst the same resistance was demonstrated in osteosarcoma samples via miR-153-3p/ABCB1 axis [170]. Surprisingly, linc-ROR expression increased the sensitivity of lung adenocarcinoma cells to cisplatin by targeting the PI3K/Akt/mTOR signaling pathway [88].

The function of linc-ROR as a promoter of resistance to chemotherapeutics through various pathways in breast cancer has also been investigated. Zhou et al. [70] investigated the breast cancer cell line MCF-7 along with tumor tissue samples to show that linc-ROR promotes the survival of breast cancer cells during rapamycin treatment by functioning as a ceRNA sponge for miR-194-3p, which targets the methyl-CpG binding protein 2 (MECP2). Another study indicated that in natural tamoxifen-resistant breast cancer cells (MDA-MB-231), the downregulation of linc-ROR could inhibit EMT and enhance sensibility to tamoxifen by increasing miR-205 expression and suppressing ZEB1 and ZEB2 [69]. Furthermore, the same cell line observed decreased sensibility to 5-FU and paclitaxel with decreased E-cadherin expression, and increased vimentin and N-cadherin expression [90], whereas yet another study with the same cell line showed that linc-ROR suppresses gemcitabine-induced autophagy and apoptosis by silencing miR-34a expression [66]. Moreover, linc-ROR inhibition reversed resistance to tamoxifen by inducing autophagy in the breast cancer cell line BT474 [91].

Active research is currently underway to link linc-ROR chemotherapy resistance in hepatocellular carcinoma. For instance, linc-ROR promoted HCC resistance to docetaxel by inducing EMT via interacting with TWIST1 in a cell model [124], while arsenic trioxide resistance was conferred to liver cancer cells through inhibiting p53 expression by linc-ROR [129]. The same linc-ROR/p53 chemoresistance pathway was demonstrated in NPC tissue specimens and cell lines [140]. Interestingly, linc-ROR was transcriptionally activated by the forkhead box M1 (FOXM1) gene, whereas the sponging of miR-876-5p by linc-ROR released FOXM1, forming a positive-feedback loop and impairing sensitivity to sorafenib in HCC cells [126]. Another experiment with sorafenib showed a positive correlation between sorafenib and linc-ROR expression in both tumor cells and extracellular vesicles, exhibiting linc-ROR-dependent effects on tumor-initiating cells and chemoresistance [127].

In gastric cancer cells resistant to Adriamycin (ADR) and vincristine (VCR), linc-ROR was upregulated, and with its depletion, reduced MRPI expression and increased apoptosis of drug-resistant cancer cells [115]. Another study demonstrated that linc-ROR could enhance the stemness features of EML-ALK+ non-small-cell lung cancer, and the expression of linc-ROR was inhibited because of the increased concentration of crizotinib [120]. Known molecular axes that promote chemoresistance and are related to linc-ROR expression are shown in Figure 3.

## 7. Clinical Relevance of Linc-ROR in Cancer

Although it is of crucial importance to understand the molecular pathways by which linc-ROR exerts its positive and negative effects in cancer, the clinicopathological implications must be widely explored to offer practical diagnostic, prognostic, and treatment alternatives. In this regard, many efforts have been conducted to uncover the relationship between the presence of linc-ROR and the most prevalent cancer types. Different studies searched databases to perform two meta-analyses in which a positive correlation in various cancer types was made between linc-ROR overexpression and several cancer features such as a more advanced clinical stage, earlier tumor metastasis, lymph node metastasis, and vascular invasion, demonstrating a very possible function as a proto-oncogene [199,200].

In colorectal cancer, 52 tissue samples were analyzed to observe a positive correlation between linc-ROR expression and tumor size, lymph node involvement, and distant metastasis [147]. Similar results were found by Zhou et al. [150] in 152 colon cancer tissue samples, where patients in the high linc-ROR expression group had significantly poorer outcomes than those of the low linc-ROR expression group. Moreover, linc-ROR overexpression was associated with poorer survival in a CRC study that analyzed 24 tumor tissues and paired normal tissue adjacent to the tumor [152]. A different approach was carried out in 351 paraffin-embedded blocks of colon cancer tissue specimens to identify the presence of genomic variants of linc-ROR; the rs1942347*A allele variant was associated with high pathological grade, larger tumor size, distant metastasis, and mortality [148]. Another study including 968 participants newly diagnosed with breast cancer studied the presence and possible correlation between functional linc-ROR single nucleotide polymorphisms (SNPs) and the disease. The results showed that interactions of SNPs in linc-ROR and reproductive factors might contribute to BC risk, and alleles of rs4801078 might affect the linc-ROR expression level [95]. Interestingly, 69 breast cancer samples and paired non-cancerous samples were examined and the relative expression of linc-ROR in tumoral tissues compared with non-tumoral tissues was associated with a history of hormone replacement therapy [98].

Intriguingly, linc-ROR expression in gastric cancer has evidenced mixed results. In 105 paired GC tissues and adjacent normal tissues, the linc-ROR expression level was significantly decreased in GC tissues compared with its adjacent nontumor tissues, and the overall survival rate of GC patients with high expression of ROR was significantly higher than those with low expression [86], whereas Liu et al. [113] related linc-ROR expression level to tumor grade, lymph node metastasis, and TNM stage in GC tissues. Conversely, the expression levels of linc-ROR- HOXA-AS2, and MEG2 were lower in gastric cancer samples compared to non-cancerous samples. Interestingly, expressions of linc-ROR and HOXA-AS2 were not associated with any clinical or pathological parameter [116].

The clinical relevance of linc-ROR in other cancers has also been reported. In NSCLC the higher linc-ROR expression levels were positively correlated with advanced TNM stage, positive distant metastasis, lymph node metastasis, and poor prognosis [119,121]. Similarly, the linc-ROR expression level was significantly associated with tumor sizes and lymph node metastasis in gallbladder cancer [191], whereas, in pancreatic cancer, linc-ROR upregulation was related to poor prognosis [134]. Furthermore, the survival rate of ESCC patients with high linc-ROR expression was lower in comparison with a low ROR expression [145], whilst linc-ROR high expression in renal cancer was associated with poor prognosis in terms of tumor undifferentiation, lymph node infiltration, postoperative recurrence, and shorter overall survival [156,158]. In addition, the higher expression of linc-ROR was associated with poor disease progression-free and overall survival as well as a younger age of patients with glioblastoma [165], whereas elevated linc-ROR was closely correlated with advanced TNM stage, lymph node metastasis, and poor overall survival rate in 48 osteoblastoma patients [172].

Several studies have discovered the overexpression of certain lncRNAs in the body fluids of cancer patients, mainly the plasma [201,202,203]. Two studies compared plasma samples by qRT-PCR, revealing linc-ROR expression significantly higher in the plasma of breast cancer patients, a positive correlation with lymph node metastasis, and a negative relation in postoperative samples [93,94]. Another study analyzed several lncRNAs in 155 women samples (50 preoperative, 40 non-cancers with risk factors, and 65 normal, non-risky volunteers) to effectively associate linc-ROR expression with the metastatic potential of breast cancer [103]. Similarly, linc-ROR expression in the plasma was closely related to FIGO stage, tumor grade, and lymph node metastasis in 60 cases of ovarian cancer patients [106]. Conversely, linc-ROR was significantly lower in NPC patients, establishing a correlation with positive Epstein–Barr virus DNA [141], whereas individuals carrying the linc-ROR rs2027701 with one or two variant alleles had significant associations with reduced risk of neutropenia [204].

## 8. Conclusions

Recently, lncRNAs have been related to the mechanisms of initiation, progression, metastasis, and chemoresistance in cancer, which has generated active research to try to elucidate the possible pathways by which they contribute to the molecular complexity of cancer. Although linc-ROR is predominantly documented as an onco-lncRNA, evidence persists in some types of tumors that contradict this function and position linc-ROR as a tumor suppressor. This review provides current synthesized information on linc-ROR molecular behavior, and portrays the clinical relevance known to date in an effort to elucidate further mechanisms to help improve cancer molecular behavior. Even though there are a considerable number of publications linking linc-ROR with various types of cancer, further investigation is needed to potentially develop useful diagnostic and prognostic tools alongside targeted and specific therapies.

## Figures and Tables

**Figure 1 ncrna-09-00012-f001:**
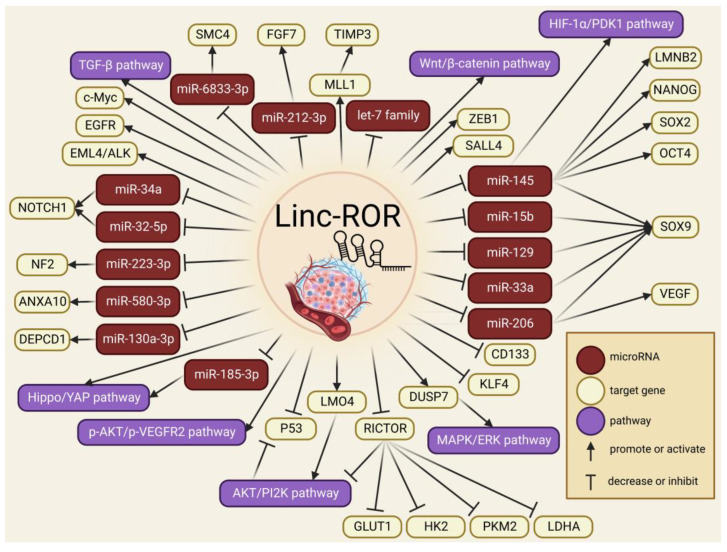
Overview of the molecular mechanisms by which linc-ROR participates in cancer proliferation and progression.

**Figure 2 ncrna-09-00012-f002:**
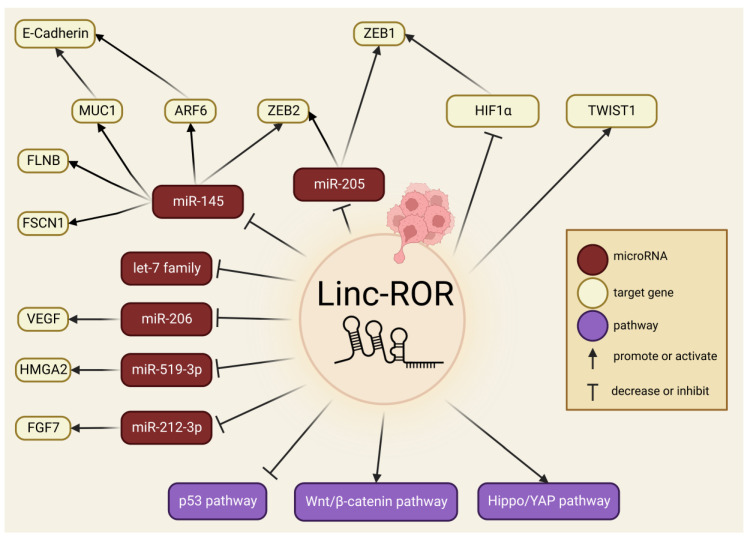
Overview of the molecular landscape by which linc-ROR participates in EMT, cancer invasion and metastasis.

**Figure 3 ncrna-09-00012-f003:**
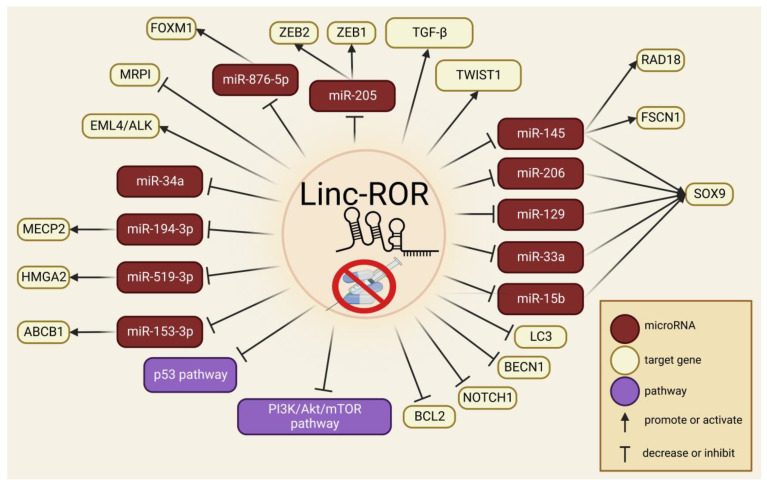
Overview of the molecular mechanisms by which linc-ROR participates in cancer drug resistance.

**Table 1 ncrna-09-00012-t001:** Functional and regulatory mechanisms of linc-ROR in cancer.

Type of Cancer	Target/Relation	Effect	Reference
Breast	MAPK/ERK pathway	Promotes estrogen-independent proliferation	[89]
	miR-194-3p	Promotes rapamycin resistance	[70]
	N- and E-cadherin, vimentin	Promotes 5-FU and paclitaxel resistance and EMT	[90]
	miR-205, ZEB1, ZEB2	Promotes tamoxifen resistance and EMT	[69]
	LC2, Beclin 1	Promotes tamoxifen resistance by autophagy	[91]
	miR-205, ZEB2	Promotes EMT	[92]
	Estrogen and progesterone receptors	Promotes lymph node metastasis	[93,94]
	Reproductive factors	Higher risk	[95]
	hnRNPI, AUF1	Promotes proliferation and tumorigenesis	[96]
	MLL1/H3K4/TIMP3	Promotes progression	[97]
	CTBP1-AS2, SPRY4-IT1	Promotes pathogenesis	[98]
	miR-145/ARF6	Promotes metastasis and invasion	[99]
	miR-145/MUC1/E-cadherin	Promotes metastasis and invasion	[100]
	TGF-β pathway	Promotes proliferation and invasion	[101]
	miR-34a	Promotes autophagy and gemcitabine resistance	[66]
	Wnt/β-catenin pathway	Promotes viability, migration, and invasion	[102]
	ND	Promotes metastasis	[103]
Ovarian and Endometrial	Wnt/β-catenin pathway	Promotes EMT and metastasis	[104]
	ND	Promotes proliferation, invasion, and metastasis	[105]
	CA125	Promotes lymph node metastasis	[106]
	miR-145/FLNB	Promotes EMT and invasion	[107]
	miR-145	Promotes stemness	[108]
	miR-34a, Notch	Promotes proliferation and suppresses apoptosis	[109]
Gastric	SALL4	Promotes maintenance and aggressiveness	[110]
	miR-519d-3p/HMGA2	Promotes proliferation, EMT and cisplatin resistance	[111]
	miR-212-3p/FGF7	Promotes proliferation, migration, and invasion	[112]
	Vimentin, E-cadherin, β-catenin, c-Myc	Promotes EMT and lymph node metastasis	[113]
	OCT4, SOX2, NANOG, CD133	Promotes proliferation and invasion	[114]
	MRPI	Promotes Adriamycin and vincristine resistance	[115]
	ADAR, FUS	Increased survival rate	[86]
	HOXA-AS1	Downregulated	[116]
	miR-580-3p/ANXA10	Suppresses proliferation, migration, and invasion	[117]
Lung	miR-145/FSCN1	Promotes docetaxel resistance	[118]
	NA	Promotes distant and lymph node metastasis	[119]
	EML4-ALK	Promotes stemness and crizotinib resistance	[120]
	P53/miR-145	Promotes proliferation, migration, and invasion	[121]
	PI3K/Akt/mTOR	Suppresses cisplatin resistance	[88]
Liver	miR-145/ZEB2	Promotes EMT and metastasis	[25]
	miR-145/RAD18	Promotes radioresistance	[122]
	IL-1β	Promotes release of pro-inflammatory cytokines	[123]
	E-cadherin, vimentin, TWIST1	Promotes EMT and Adriamycin resistance	[124]
	DEPCD1	Promotes progression and angiogenesis	[125]
	miR-876-5p/FOXM1	Promotes sorafenib resistance	[126]
	TGF-β	Promotes sorafenib resistance	[127]
	miR-145/HIF-1α	Promotes survival during hypoxic stress	[128]
	P53	Promotes arsenic trioxide resistance	[129]
	miR-223-3p/NF2	Promotes proliferation and invasion	[130]
	OCT4, NANOG, SOX2, p53, CD133	Promotes proliferation	[131]
Pancreatic	ZEB1	Promotes EMT and aggressiveness	[132]
	Hippo/YAP pathway	Promotes EMT, proliferation, and invasion	[24]
	HIF1-α/ZEB1	Promotes EMT	[133]
	miR-145, NANOG	Promotes proliferation and decreases migration	[68]
	Let-7 family	Promotes migration, invasion, and EMT	[134]
Head and Neck	miR-145-5p	Promotes stemness	[135]
	ND	Promotes progression and metastasis	[136]
	LMO4/AKT/PI3K	Promotes proliferation and invasion	[137]
	p-AKT/p-VEGFR2	Promotes proliferation, migration, and angiogenesis	[138]
	P53	Promotes proliferation, metastasis and inhibits apoptosis	[139,140]
	ND	Downregulated in plasma	[141]
	P53		
Esophageal	miR-15b, miR33a, miR-129, miR-145, miR-206	Promotes proliferation, motility, chemoresistance, and renewal capacity	[142]
	ND	Promotes initiation and progression	[143]
	miR-204-5p/MDM2/p53	Suppresses apoptosis	[144]
	miR-145/FSCNI	Promotes metastasis	[145]
	miR-145/LMNB2	Promotes proliferation and migration	[146]
Colorectal	miR-145	Promotes stemness and metastasis	[147]
	hnRNPI, AUF1	Promotes proliferation and tumorigenesis	[96]
	NA	Related with larger tumor size, metastasis, and mortality	[148]
	P53/miR-145	Promotes radioresistance and suppresses apoptosis	[149]
	miR-145	Promotes lower survival rate	[150]
	EGFR	Promotes proliferation invasion, and migration	[151]
	miR-6833-3p/SMC4	Promotes proliferation and lower survival rate	[152]
	P53	Promotes proliferation and viability	[153,154]
	CCAT1	Promotes metastasis	[155]
Kidney and Bladder	SOX2, Nanog, POU5F1	Promotes stemness, infiltration and shorter survival	[156]
	ZEB1	Promotes proliferation, metastasis, EMT, and inhibits apoptosis	[157]
	P53, c-Myc	Promotes shorter survival rate	[158]
	miR-206/VEGF	Promotes proliferation and metastasis	[159]
	TESC	Promotes tumorigenesis	[160]
Thyroid and Parathyroid	TESC/ALDH1A1/ TUBB3/PTEN	Promotes progression	[161]
	miR-145	Promotes EMT	[162]
	ND	Promotes EMT and metastasis	[163]
	ND	Suppresses progression	[164]
Brain and Retina	ND	Promotes poor overall survival	[165]
	EGFR	Promotes proliferation and stemness	[166]
	KLF4/CD133	Suppresses proliferation	[87]
	ND	Promotes proliferation and angiogenesis	[167]
	Akt pathway	Suppresses proliferation	[168]
	miR-32-5p/Notch	Promotes EMT, invasion and metastasis	[169]
Bone	miR-153-3p/ABCB1	Promotes cisplatin resistance	[170]
	miR-185-3p/YAP1	Promotes growth and metastasis	[171]
	miR-206	Relates to advanced TNM, metastasis and poor survival	[172]
Skin	P53, PI3K/Akt	Promotes proliferation	[173]
Prostate	miR-145/Oct4	Promotes proliferation, invasion, and tumorigenicity	[174]

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
