# Peer review of "Functional Relevance of the Long Intergenic Non-Coding RNA Regulator of Reprogramming (Linc-ROR) in Cancer Proliferation, Metastasis, and Drug Resistance"

_ncrna, 2023, doi:10.3390/ncrna9010012_

Round 1
Reviewer 1 Report
This review manuscript aims at providing comprehensive information on linc-ROR in different pathological conditions. The number of publications considered is huge and the authors make a very detailed work on it.
I suggest to extend what reported in section 3 including a description of linc-ROR structure, quantitative distribution, localization and sites of interaction with miRNA if known (maybe adding a figure/diagram). This information provided before the list of the findings in the different considered disorders could help the reader to figure out the contribution of this non coding RNA to the alteration of the pathways involved which would otherwise result very descriptive.
minor points:
page 3, line 113 "and they constitute more than half of lncRNA transcripts in humans". are the authors referring to the absolute amount (copies) of the number of different transcripts, please specify.
Author Response
The authors would like to thank the reviewer for reading the paper and providing kind suggestions to improve the manuscript. We look forward to keep working with your expertise.
Please see the attachment for a proper response to the points suggested.
Best regards,
José A. Peña-Flores

Reviewer 2 Report
The manuscript by Peña-Flores et al. (ncrna-2191885) reviews the involvement of long intergenic non-coding RNAs, specifically the regulator of reprogramming (linc-ROR) in cancer tumorigenesis, progression, metastasis, and chemoresistance.
This review is mostly clearly-written and provides relevant and logically structured information about this interesting field, although do not help in the understanding of the molecular pathways which underlies the contribution of long ncRNA in cancer in general.
I hope that my following comments can help to get an improved version:
- Clinical relevance does not seem to be a main aspect of the manuscript, but secondary. Therefore, consider either adapting the title or more in-depth on this throughout the Review, not just in the last section.
- Not clear about the aim in the Abstract. Please, rewrite lines 11-19
- Given the large amount of material already reviewed and published on this topic, it is unclear the significant contribution of this review to the field. I suggest the author highlight this point in the Abstract, Introduction, and Conclusion sections.
- I found vague the introduction of cancer and metastasis in 1., mainly focused on RNA. I missed some descriptions similar to those made in lines 169-172.
- Lines 57-60, and 73-75: could you further explain the text?
Author Response
The authors kindly thank the reviewer for reading the manuscript and providing feedback to improve the paper. We are sorry to read that it is the reviewer's opinion that our review does not help in the understanding of the molecular pathways, as it is the author's opinion that the review makes an up-to-date synthesis of linc-ROR's participation in diverse cancer mechanisms. We appreciate and value every thought and opinion given, and we'd like to thank you again for your time and effort, as we will look forward to work with your expertise in the future.
Please see the attachment for a proper response to the points you kindly mentioned.
Best regards,
José A. Peña-Flores
